# An Overview of Effects Induced by Pasteurization and High-Power Ultrasound Treatment on the Quality of Red Grape Juice

**DOI:** 10.3390/molecules25071669

**Published:** 2020-04-04

**Authors:** Alina Margean, Mirabela Ioana Lupu, Ersilia Alexa, Vasile Padureanu, Cristina Maria Canja, Ileana Cocan, Monica Negrea, Gavrila Calefariu, Mariana-Atena Poiana

**Affiliations:** 1Food and Tourism Faculty, Transilvania University of Brasov, Castelului 148, 500014 Brasov, Romania; alinamargean@yahoo.com (A.M.); padu@unitbv.ro (V.P.); canja.c@unitbv.ro (C.M.C.); 2Faculty of Food Engineering, Banat’s University of Agricultural Sciences and Veterinary Medicine “King Michael I of Romania” from Timisoara, Calea Aradului 119, 300641 Timisoara, Romania; alexa.ersilia@yahoo.ro (E.A.); ileanacocan@usab-tm.ro (I.C.); negrea_monica2000@yahoo.com (M.N.); atenapoiana@yahoo.com (M.-A.P.); 3Faculty of Technological Engineering and Industrial Management, Transilvania University of Brasov, Universitatii 1, 500068 Brasov, Romania; gcalefariu@unitbv.ro

**Keywords:** red grape juice, pasteurization, high-power ultrasound treatment, phenolic compounds, microorganisms

## Abstract

In juice processing, ultrasound treatment has been tested as a potential alternative to conventional thermal methods to inactivate microorganisms and to enhance the nutritional status of juice. In this study, the impact of pasteurization and high-power ultrasound treatment on the quality of red grape juice was investigated in terms of the content of bioactive compounds such as phenolic compounds and l-ascorbic acid as well as regarding the microbiological and physicochemical properties. The grape juice was subjected to pasteurization (80 °C, 2 min) as well as to ultrasound treatment with an amplitude of 50 and 70% for 5 and 10 min. The results indicated the same level of total phenolic content for pasteurized and sonicated samples for 10 min with an amplitude of 70%, while the highest level of l-ascorbic acid was recorded for sonicated samples with an amplitude of 70% for 10 min. pH of sonicated samples decreased with amplitude and treatment time while total soluble solids and titratable acidity increased with amplitude and time. Moreover, the results indicated the usefulness of juice sonication to enhance the inactivation of microorganisms. Thus, the high-power ultrasound treatment might represent a viable technique to replace the conventional thermal treatment in grape juice processing.

## 1. Introduction

Grapes are one of the largest fruit crops harvested around the world. However, grape juice is not consumed worldwide in large amounts because of its higher sugar content or higher acidity, but it can be mixed with apple or other juices [1,2,3]. The last decades were characterized by numerous studies related to the chemical constituents of grapes, especially phenolic compounds that are secondary plant metabolites with free radical scavenging activity [4,5]. The main phenols in grape belong either to the flavonoid family or the non-flavonoid one. Among the flavonoid family, anthocyanins, flavanols, flavonols and dihydroflavonols are considered to play an important role in the pigmentation and flavor of grapes berries, while non-flavonoid phenolic compounds such as resveratrol have a role in the powerful biological activities of grape [6,7].

Many research studies have suggested that antioxidants might have positive effects on human health due to their effects against oxidative stress. Thus, it was considered that a diet based on a high daily intake of antioxidant dietary supplements from fruits and vegetables might result in a lower incidence of atherosclerosis, Alzheimer’s disease, cancer, ocular disease, diabetes, rheumatoid arthritis and motor neuron disease [8].

In order to reduce microbial grow and to extend their shelf-life, fruit juices are usually processed by pasteurization. Besides the fact that this thermal procedure influences the quality of the products, it also has the disadvantage of denaturing the bioactive compounds due to ionization, hydrolysis and oxidation reactions [5,9]. Considering consumers’ need for high quality, flavor and taste of fruit juices, producers are continuously looking for alternatives to the conventional thermal methods. Moreover, the consumers demand for nutritious and safe food products has resulted in increasing interest in non-thermal preservation techniques.

Recently, emerging technologies like ultrasound, microwave and pulsed electric field irradiation have been tested in the food industry to develop various effective food processing applications [10]. Ultrasounds are sound waves of frequencies higher than 16 kHz (higher than usually detected by the human ear). When transmitted through a liquid, solid or gas with elastic properties, the sound waves are moving at a speed depending on the wavelength and the type of material. When acting on a liquid medium, the fundamental effect of ultrasounds is to add to the hydrostatic pressure (P_h_) an acoustic pressure (P_a_) dependent on time (t), wave frequency (f) and the maximum amplitude of wave P_A_. The relationship among the above factors is illustrated by the Equation (1) [10]:P_a_ = P_A_ × sin(2 × π × f × t)(1)

The propagation of ultrasound waves in a medium generates physical and chemical effects which can be exploited for improving the efficiency of various food processing operations. Thus, the ultrasound technology was explored to enhance the food quality by reducing the process time, energy and increasing shelf life [11,12]. It has been reported that ultrasound energy can be used to increase extraction yields by disrupting cell tissues [12,13].

In terms of their applications, two approaches are usually used for ultrasound. Low-power ultrasound (LPU), at frequencies above 100 kHz, is a non-invasive technique and might be applied to monitor food processes and to evaluate the physico-chemical food properties, while high-power ultrasound (HPU), with frequencies between 18–100 kHz, causes physical, mechanical and chemical effects and might offer insight into the long term stability of fruit juices [14]. Most often, the power ultrasound range is used in the extraction of biologically active compounds from biological matrices, such as in the extraction of polyphenols from plant and waste food materials. Most of the commercial ultrasonic probes or cleaning baths work at low ultrasound frequencies (20–40 kHz) [15]. Low frequencies have been observed to generate large cavitation bubbles in extraction solvents which implode violently, generating high shear whiles creating microjets that ensure higher cellular degradation, increased solvent penetration and higher extraction rates [15,16]. In juice processing, HPU treatment might be used not only for controlling spoilage microorganisms, but also as an efficient and environmentally friendly treatment for enhancing the nutritional status of juices in terms of increasing the bioactive compounds [7,17]. It was reported that ultrasound was applied in extraction of plant materials because of increased sugar content, total acid content, phenolics content as well as color density of grape juice [2]. Moreover, the ultrasound treatment can influence the quality parameters of cranberry juice and nectar in terms of aromatic profile and sensory properties [18], the stabilization of cantaloupe melon juice [19] or the preservation of monomeric anthocyanins along with a significant microbial load reduction during storage [20].

Therefore, our study represents an attempt to identity alternative fruit processing techniques in order to reduce the microbial load and to improve the nutritional quality of the obtained products. Moreover, this research reflects the possibility to replace the conventional thermal treatment with HPU treatment as a potential tool for setting up a more efficient and sustainable technology for juice processing.

In line with the above-mentioned considerations, the goal of this research was to investigate the impact of pasteurization and HPU treatment on the quality of red grape juice. In this purpose, the content of bioactive compounds such as phenolic compounds and l-ascorbic acid as well as the physicochemical attributes and the microbiological properties of pasteurized grape juice samples were compared with those of the juice samples subjected to HPU treatments at different parameters such as amplitudes of 50% (A50%) and 70% (A70%) for 5 and 10 min.

## 2. Results and Discussion

The effects of pasteurization and high-power ultrasound (HPU) treatments on the quality of red grape juice are discussed in terms of bioactive compounds such as total phenolic (TP) content, l-ascorbic acid (L-AsAc) or vitamin C content and the profile of polyphenolic compounds. Also, the changes in physicochemical parameters such as pH, total soluble solids (TSS) and titratable acidity (TA) as well as the microbiological quality expressed by total plate counts (TPC) and *Enterobacteriaceae* count (ENT) were investigated in response to the applied treatments. All mentioned parameters were investigated for untreated red grape juice, as a control sample (C), as well as for red grape juice samples subjected to pasteurization at 80 °C for 2 min (P), respectively to sonication with an amplitudes of 50 and 70% for 5 and 10 min, as follows: HPU (A50%, 5 min), HPU (A50%, 10 min), HPU (A70%, 5 min) and HPU (A70%, 5 min). Some authors also found that ultrasound processing parameters like amplitude and time plays a decisive role in both extraction and retaining of bioactive compounds and in determining the effects on the quality attributes [21].

To provide an overview of the effects induced by pasteurization and HPU treatments on investigated parameters, compared to untreated grape juice samples, the results obtained in this study were processed by a one-way ANOVA test. Based on the data obtained through statistical processing the significance of changes recorded in the investigated parameters in response to the applied treatments, reported with respect to the control grape juice sample can be determined. Also, the changes occurring in the investigated parameters as a result of HPU treatments were evaluated relative to pasteurization for assessing the possibility of replacing the conventional thermal treatment currently used in the grape juice processing, with HPU treatment, as a non-thermal food processing technique.

### 2.1. Effect of Operating Parameters of HPU Treatments on Temperature of Grape Juice Samples

The operating parameters used during the HPU treatments can impact the temperature of the grape juice samples. The average temperature values measured every 2.5 min for ultrasound- treated juice samples are shown in Table 1 while in the Figure 1 and Figure 2 the values measured by a forward-looking infrared (FLIR) camera at the end of sonication treatment with 50 and 70% amplitude can be seen.

During ultrasound treatment, additional attention was paid to measure the heat generated in the samples as an effect of the sonication. To account for the heat issue and the corresponding increase of the temperature, a thermovision camera was used to ensure the temperature of the sample did not exceed 60–66 °C. Temperatures above 70 °C in particular have been shown to lead to rapid polyphenol degradation, hence the need to select efficient extraction temperatures that ensure the stability of phenolic compounds [15,22,23]. It is also important to mention that the sensitivity of a sample to temperature-induced polyphenol degradation depends on the types of polyphenol compounds available in the extract or plant matrix, and their physicochemical and biochemical characteristics, as well as solvent-sample interactions [15].

Preliminary studies related to the temperature increase mention that at a power of 100 W and a frequency of 20 kHz, a treatment over 15 min in duration resulted in a temperature in the sample of about 50 °C [21].

The results obtained by Lieu and Le revealed that temperature and time had positive effects on yield of the treatment process [2]. We observed that the pattern of the average temperatures increasing was linear for both 50 and 70% amplitudes. When the higher amplitude was applied, i.e. 70%, the average temperatures increased by 13.7 °C for the same sonication treatment of 5 min and by 15.8 °C after 10 min.

### 2.2. Effects of Pasteurization and HPU Treatments on Total Phenolic (TP) Content, l-ascorbic Acid (L-Asac) Content and Polyphenolic Compounds Profile

In Table 2 the total phenolic (TP) and l-ascorbic acid (L-AsAc) contents of red grape juice samples after pasteurization and HPU treatments are presented. These results reveal decreases in TP content, with respect to the control sample, in response to pasteurization and ultrasound treatments.

The content of total phenols quantified as gallic acid equivalent (GAE) in treated red grape juice samples varies from 50.68 to 70.64 mg GAE/100 mL. Another study reported a level of TP content in red grape juices in the range 744–1177 mg GAE/L [24]. TP content among sonicated samples was significantly affected by the amplitude level (*p* < 0.05) and not impacted by the treatment time (*p* > 0.05).

In Figure 3 the changes of TP content in juice samples in response to the applied treatments are depicted. It can be noted that there were losses in the TP content of the grape juice samples in response to pasteurization compared to ultrasound treatment. Thus, Figure 3a shows the losses of TP content induced by pasteurization and HPU treatment with respect to the control sample while Figure 3b reveals the losses in juice sampled subjected to HPU treatment compared to the pasteurized sample.

The first thing we could notice about the data from Figure 3a is that the pasteurization induced the lowest decreases in TP content, compared to the control sample (3.05%), while the HPU treatments (A50%, 5 min; A70%, 5 min and A50%, 10 min) led to the losses in the range 25–30%. It is worth noting that the HPU treatment (A70%, 10 min) has an effect close to that obtained through pasteurization, as regards the TP content losses (4.09%).

A closer look at Figure 3b reveals that that the losses of TP content induced by HPU treatments (A50%, 5 min; A70%, 5 min and A50%, 10 min) were in the range of about 22–28%, compared to the pasteurized sample. The HPU treatment (A70%, 10 min) induced the lowest losses in TP content, about 1% compared to the pasteurized sample. Therefore, by choosing an HPU treatment for 10 min with an amplitude of 70%, the retention of total phenolic compounds in grape juice sample could be improved.

The initial content of l-ascorbic acid in untreated red grape juice sample was 454.4 mg/L. Decreases in the l-ascorbic acid content in response to pasteurization and HPU treatment applied to red grape juice were recorded, indicating that ascorbic acid is highly thermosensitive. The l-ascorbic acid content in grape juice samples decreased by 25% in response to pasteurization and HPU treatment (A50%, 5 min) and by 53% by applying HPU treatment at a higher amplitude (A70%, 5 min).

The degradation of l-ascorbic acid is probably attributable to the oxidation processes that occur during these treatments. The results were in agreement with other previous studies. Thus, data reported by Cao et al. [25] reveals an initial content of ascorbic acid in fresh bayberry juice of 22.82 mg/100 mL and a decrease of 25.16% in response to pasteurization. Adekunte et al. [26] also reported that the content of ascorbic acid of tomato juice decreased by 96.9 to 60.7% in response to ultrasound treatment of 24.4–61 μm for 2–10 min.

However, after HPU treatment (A70%, 10 min) an increase in L-AsAc content to 568 mg/L in comparison with the untreated grape juice was recorded. An increase of ascorbic acid was also detected by other authors at high amplitude [27,28,29]. An explanation could be the elimination of dissolved oxygen, essential for ascorbic acid degradation, during the cavitation produced by the sonication treatment [28] which prevents ascorbic acid breakdown.

Figure 4 provides information regarding the changes recorded in L-AsAc content in juice samples in response to the applied treatments. It can be observed that by exposing the grape juice to pasteurization, compared to ultrasound treatments, both losses and increases in the content of L-AsAc were recorded. The effect of P and HPU treatment on L-AsAc content reported to the control sample is shown in the Figure 4a. In addition, the changes in L-AsAc content recorded in juice samples subjected to HPU treatment versus the pasteurized sample are presented in Figure 4b.

Our data reveal that pasteurization and the HPU treatments (A50%, 5 min; A70%, 5 min) induced losses in L-AsAc content as follows: 25, 50 and 23.02% with respect to the control sample. Contrary to this finding, by applying HPU treatments (A50%, 10 min; A70%, 10 min) increases in L-AsAc content of 0.88, or 25%, were recorded compared to the control sample (Figure 4a). As regards the changes in L-AsAc content induced by HPU treatment with regards to the pasteurized sample, the data depicted in Figure 4b prove that only by HPU treatment (A50%, 5 min) were losses observed. By applying the other HPU treatments (A70%, 5 min; A50%, 10 min; A70%, 10 min) increases in L-AsAc content, compared to the pasteurized sample, were recorded as follows: 2.64, 34.51 and 66.67%,. These data highlight that the HPU treatment (A70%, 10 min) was the most effective among the investigated treatments to ensure a high content of vitamin C in grape juice samples.

Polyphenols play an important role among red grape components as they determine the flavor and color of the juice [30,31] and have several beneficial properties for human health [30,32]. The evolution of individual phenolic compounds of red grape juice is shown in Table 3. Among phenolic compounds, the most known is resveratrol. Significantly higher amounts of resveratrol were observed in the grape juice samples exposed to ultrasound treatment, especially at A70%, 5 min (8.54 mg/L). Resveratrol was not detected in the juice sample subjected to pasteurization that suggests that during this thermal treatment it was destroyed. An increase of resveratrol in the ultrasound processing of grape juice was also observed by Hasan et al. [33]. A slight increase in caffeic acid has been reported in the clear apple juice treated with ultra-high pressure homogenization which is also a non-thermal technique [34] and also in a study on apple juice treatment by ultrasound at 70% amplitude and 25 kHz frequency [35]. An increase in gallic acid content from 2.15 mg/L recorded in the control sample to 2.16 mg/L in pasteurized samples and to 31.3 mg/L in a juice sample subjected to HPU treatment (A70%, 5 min) can be noted The highest level of protocatechuic acid (5.13 mg/L), epicatechin (76.80 mg/L), *p*-coumaric acid (1.20 mg/L), rutin (63.31 mg/L), rosmarinic acid (25.93 mg/L), quercetin (5.62 mg/L) and kaempferol (4.59 mg/L) were also detected in the grape juice samples subjected to HPU treatment (A70%, 5 min). The phenolic compounds can be found both in soluble form in the vacuole or bound to the other compounds such as cellulose, hemicellulose, pectin and lignin traces of the cell wall [28]. The increase of polyphenolic compounds in sonicated juice samples might be ascribed to the increase in the extraction efficacy by ultrasounds treatment causing disruption of cell walls and ultimately the liberation of bound polyphenolic compounds [27,36]. Also, the ultrasound treatment might enhance the release of phenolic compounds from the cell walls, due to the collapse through cavitation process in the surroundings of colloidal particles found in juice [28].

We can appreciate that the overall polyphenolic profile of the red grape juice samples is given by the contribution of a multitude of individual polyphenolic compounds. It is well known that red grapes are recognized for the significant amount of anthocyanins in their composition (300–7500 mg/kg fresh weight) and less for the content of monomeric flavanols such as epicatechin (30–175 mg/kg fresh weight) or flavonols as quercetin or kaempferol (15–40 mg/kg fresh weight) [37]. Antocyanins, the most relevant polyphenolic compounds in red grapes, with flavonoid-like glycoside structures, show similar behavior under heating and their degradation depends on the temperature and pH [38]. At pH = 1 all the glycosidic bonds are susceptible to hydrolysis, while at a pH values in the range 2–4, specific for grape juice, aglycone-sugar bond is the most labile of the glycosidic bonds [38]. This means that under our working conditions the anthocyanins are less susceptible to destruction compared to flavonoids, which might justify the tendency of total phenolic content to increase, compared to the investigated individual polyphenolic compounds. In addition, the increase of total polyphenols content can also be attributed to hydroxycinnamic acids whose content can increase with the amplitude, respectively with the temperature.

The increase of HPU treatment time from 5 to 10 min, for a constant amplitude of 50 and 70%, leads to a significant increase in the temperature of the process, as was shown in Figure 1 and Figure 2. The HPU treatment (A70%, 5 min) produces a heating from 23.1 up to 66.2 °C with an average value of 50.1 °C, while for HPU (A70%, 10 min) the temperature increases from 23.9 up to 85.5 °C with an average value of 66 °C. At a higher temperature, exceeding 60 °C, especially over extended periods, some phenolic compounds suffer oxidative degradation. It is also important to mention that the sensitivity of a sample to temperature-induced polyphenol degradation depends on the types of available polyphenol compounds [15]. Sharma et al. [39], highlighted that after heating, a decrease in total flavonoids was observed, which indicates that some flavonoids were probably destroyed, however, the total phenolics were increased [39]. In most fruits and vegetables, flavonoids contain C-glycoside bonds and exist as dimers and oligomers. Thermal food processing methods such as heating or boiling result in the formation of monomers by the hydrolysis of C-glycosides bonds [39]. Ross et al. [40], have shown that catechin and epicatechin content decreased with increasing heating temperature, but the content of some hydroxycinnamic acids increased.

According to the reported results it can be observed that both the ultrasound processing time and amplitude are influential factors for preserving the content of active principles. Our data reveal that HPU treatment with an amplitude of 70% for 10 min led to an increase in the TP and L-AsAc content, compared to 5 min, while HPU treatment (A70%, 5 min) might favor an increase of the levels of flavonoids such as rutin, quercetin, epicatechin and kaempferol as well as resveratrol, a natural stilbene found in grape juice. At the same time, some hydroxycinnamic acids such as ferulic and caffeic acid are found in higher concentrations after exposing the grape juice to HPU treatment (A70%, 10 min).

### 2.3. Effects of Pasteurization and HPU Treatments on Ph, Total Soluble Solids (TSS) and Titratable Acidity (TA)

Physicochemical parameters such as pH, total soluble solids (TSS) and titratable acidity (TA) of untreated and pasteurized grape juice along with grape juice treated for 5 and 10 min with different ultrasound amplitudes (A50% and A70%) are shown in Table 4.

The results suggested that HPU treatment causes changes in samples as the pH decreased with increasing amplitude and treatment time. However, the pH value among sonicated samples was found to be significantly influenced by the treatment time (*p* < 0.001) and not influenced by the amplitude level (*p* > 0.05). Also, the pH values recorded for all juice samples subjected to HPU treatments were significantly different than those recorded for C and P samples (*p* < 0.001).

TSS of samples treated with different amplitudes and treatment times indicate a decrease compared with pasteurized juice. For sonicated samples, results suggested that TSS increased with amplitude and time, the value of sample sonicated with amplitude 70% for 10 min being 1.71% lower than for pasteurized juice. Although HPU treatment induced slight changes in samples, statistically the TSS content among sonicated samples was significantly influenced by amplitude level (*p* < 0.001) and not influenced by treatment time (*p* > 0.05). Also, all HPU samples were significantly different than the P sample (*p* < 0.001).

A higher value of TA was observed for pasteurized samples. For sonicated samples, results suggested that TA increased with amplitude and time, the value of sample sonicated with an amplitude of 70% for 10 min being 1.18% lower than the value recorded for the pasteurized juice sample. TA among sonicated samples was significantly influenced by both amplitude level and treatment time (*p* < 0.05).

As mentioned earlier, it is clear from the data that the different assayed combinations of ultrasound frequency and exposure time exert slight effects on the pH, TSS and TA of grape juice. Sugars are the main components of the fruit juices, being an important quality attribute and influencing consumers to accept or reject a product.

As regards the total soluble solids, around 80% are constituted by sugars [21]. The results indicated that pasteurization and sonication treatments had different effects on the total soluble solids of the grape juice. Our results related to TSS are similar to the results of sonicated sweet lime [21] and kastuni juice [41] and in contrast with the ones of sonicated orange juice and carrot [21]. However, Gao and Rupasinghe [42] obtained similar results as of both lower TSS of the sonicated sample compared with pasteurized one for an apple-carrot ratio of 60:40 and 90:10 and lower pH for an apple-carrot ratio of 90:10. Other authors found no significant changes in pH, TSS and TA due to the sonication treatment of grape juice using an ultrasonic treatment of 25 kHz at a constant temperature [27], or any changes [43] using an ultrasonic treatment of 28 kHz, respectively.

Regarding the TA, our results are not similar with the results observed by different authors in grape mash [2], apple-carrot juice [42] and grapes [44], which suggests that ultrasound treatment augmented the tartaric acid extraction. This may be attributed to the fact that we used high power ultrasound at a fixed frequency of 20 kHz.

### 2.4. Microbiological Quality

The application of sonication treatments has been recognized as a potential innovative technology that can meet Food and Drug Administration’s (FDA’s) requirements for achieving a 5-log reduction in the contaminant microorganisms (or pathogens) associated with fruit juices [45,46]. The microbiological quality of the grape juice samples was assessed on the basis of two indicators, namely the total plate count (TPC) and the *Enterobacteriaceae* count (ENT).

The total plate count (TPC) is used as a reliable indicator for the bacterial populations of a food sample. It is also called the aerobic colony count, the standard plate count, the mesophilic count or the aerobic plate count. TPC provides information about the total microbial load in a food sample. TPC quantifies the aerobic, mesophillic organisms that grow under aerobic conditions at a moderate temperature in the range 20–45 °C. The TPC includes all pathogens and non-pathogens and is used to assess the hygiene status of a food product [47]. The *Enterobacteriaceae* count (ENT) is considered by food producers as an indicator of hygiene practices, being useful to monitor the effectiveness of implemented preventive measures. The indicator microorganisms are commonly used to measure the quality of the practices used in order to ensure a proper processing of food products. The *Enterobacteriaceae* or *Escherichia coli* are used for assessing the enteric contamination of a food [48].

In Table 5 a reduction in the microbial load after pasteurization and sonication treatments of red grape juice can be observed. The control sample showed values of 4.53 log colony forming units (CFU)/mL for TPC and 1.51 log CFU/mL for ENT. The microorganisms were reduced in the pasteurized juice sample (TPC was 2.53 log CFU/mL and ENT 0.73 log CFU/mL) as well as in all samples exposed to HPU treatments. ENT and TPC of the sonicated samples were significantly influenced by the treatment time and the amplitude level (*p* < 0.001). As the data from Table 5 reveal, TPC and ENT recorded a decrease versus the control sample by increasing the amplitude level from 50 to 70% as well as by increasing the sonication time from 5 to 10 min. The lowest bacterial counts (TPC: 1.13 log CFU/mL and ENT: 0.53 log CFU/mL) were recorded after exposing the red grape juice sample to HPU treatment (A70%, 10 min). Similar results have been reported by other authors [41,45,49], revealing that the microbial load was lowered by increasing the ultrasound processing time. This indicates that the microbial cells might be resistant to sonication treatment and the cell destruction occurs only as the sonication treatment time is increased to a longer duration. Cell disruption may be caused by several factors such as combined physical and chemical mechanisms that occur during cavitation (caused by the changes in pressure), the formation of free radicals and hydrogen peroxide [49,50], leading to a thinning of microbial cell membranes, and the localized mild heating that occurs during sonication treatments [45,49].

## 3. Materials and Methods

### 3.1. Grape Juice Preparation

Fresh red grapes (Merlot, from the Pietroasa Development Research Station for Viticulture and Winery, Buzau County, Romania) were split from bunches and crushed with a crusher-destemmer (Enoventa Technologies Enologiche, Piazzola sul Brenta, Italy) followed by the pressing process with a hydraulic press machine (L.U.C.M.E. Elettromeccanica, Verona, Italy). Only fruits without external injuries were used. Juice extraction and filtration were performed in a cold room at a temperature of 10 ± 1 °C. The obtained red grape juice was further subjected to pasteurization and HPU treatments. As a control sample was used the red grape juice before applying of any treatments. The control sample as well as the pasteurized and HPU treated juice samples were kept in sterilized, airtight container (bottles) and were stored at 4 °C until further analysis. All sample preparations and treatments were carried out in triplicate [45].

### 3.2. Pasteurization Process

Pasteurization of the grape juice samples was carried out on a hotplate with a magnetic mixer (IKA RTC Basic, Ika-Werke GmbH & Co. KG, Janke & Kunkel, Staufen, Germany) where the samples (150 mL) were placed in glass containers covered with aluminium foil over hot water bath at a temperature of 80 °C for 2 min. This temperature was chosen based on the previously reported results [18].

### 3.3. Ultrasound Equipment and Processing

The ultrasound treatments were conducted using a 750 W ultrasonic processor for small and medium volume applications (VCX 750, Sonics & Materials, Inc., Newtown, CT, USA) with a 1/2″ (13 mm) probe used for ultrasound at a constant frequency of 20 kHz. A sample of 150 mL was placed in a glass beaker and the ultrasound probe was fixed at 45 mm depth in the juice sample. The amplitude was set at 50% (A50%) and 70% (A70%) and for each of these amplitudes, the grape juice samples were exposed to ultrasounds for 5 and 10 min. The main variables that are influencing the sonication are the intensity and the frequency of the waves. The amplitude of vibration of the ultrasonic source is proportional with the intensity of sonication that is the power dissipated per unit of surface area (W/cm^2^) of the sonotrode. Thus, an augmentation of the amplitude induces an increasing of the sonochemical effects in the treated liquids. In order to identify the best extraction configuration, the amplitude may vary over a wide range [10]. Thus, depending on the amplitude and duration, four ultrasounds treatments were applied to the grape juice samples, as follows: HPU (A50%, 5 min), HPU (A50%, 10 min), HPU (A70%, 5 min) and HPU (A70%, 5 min). These combinations of amplitude and time of the applied HPU treatments were established according to the results of other previous studies, where a reduction of microbial count was obtained [28,49].

### 3.4. pH, Total Soluble Solids (TSS) and Titratable Acidity (TA)

The pH was measured using the electrochemical method [51] with a potentiometer (Consort C1010, Consort, Turnhout, Belgium). Titratable acidity (TA) and total soluble solids (TSS) of red grape juice samples were determined following OIV reference methods [52]. Total soluble solids (°Brix) were measured with an ABBE refractometer (ORT 1RS, KERN & SOHN GmbH, Balingen, Germany). Titratable acidity tests were performed on aliquots of 50 mL of the sample, 30 mL of boiled distilled water and 1 mL of bromothymol blue solution placed into a 250 mL beaker and titrated with standardized 0.1 mol/L NaOH (Sigma-Aldrich, Dublin, Ireland) until the same color was obtained as in the preliminary test (end-point color determination). The results were expressed as g of tartaric acid/L of juice.

### 3.5. l-ascorbic Acid Content (L-AsAc)

l-ascorbic acid (L-AsAc) or vitamin C determination of investigated grape juice samples was carried out by the 2,6-dichloroindophenol titrimetric method [53]. For this purpose, 10 mL of each sample was diluted with 10 mL oxalic acid 2% (Sigma-Aldrich), then, the mixture was filtered through Whatman filter paper and the clear extract was used for analysis. Further, 10 mL of the clear extract was taken and 1 mL hydrochloric acid 1N (ReAgent Chemicals, Runcorn, UK) was added. The obtained mixture was titrated with 2,6-Dichloroindophenol sodium salt (Sigma-Aldrich). A control sample titration was also performed. The results were expressed as mg/L of juice.

### 3.6. Determination of Total Phenolic (TP) Content and Polyphenolic Compounds Profile

The total phenolic content was determined by Folin-Ciocalteu assay [54]. For this purpose, 0.5 mL sample was treated with 1.25 mL reagent Folin-Ciocalteu (Merck, Darmstadt, Germany) diluted 1:10 with water. The sample was incubated for 5 min at room temperature and then 1 mL Na_2_CO_3_ 60 g/L was added. After 30 min of incubation at 50 °C the absorption of samples was measured at 750 nm using a UV-VIS spectrophotometer (Specord 205, Analytic Jena, Jena, Germany). The calibration curve was prepared using gallic acid as standard in the concentration range 5-250 µg/mL. The results were expressed in mg gallic acid equivalents (GAE)/100 mL of juice.

The polyphenolic compounds profile was determinate by high performance liquid chromatography coupled with mass spectrometry (LC-MS) according to the method described by Abdel-Hameed et al. [55]. Thus, the main polyphenols from red grape juice samples were determined by LC-MS using SPD-10A UV (Shimadzu, Kyoto, Japan) and LC-MS 2010 detectors, and an EC 150/2 NUCLEODUR C18 Gravity SB 150 × 2 mm × 5 μm column (Macherey-Nagel, Düren, Germany). Chromatographic conditions were as follows: mobile phases A: water with formic acid at pH-3, B: acetonitrile with formic acid at pH-3, gradient program: 0.01–20 min 5%B, 20.01–50 min 5–40%B, 5–55 min, 40–95%B, 55–60 min 95%B. Mobile phase 0.2 mL/min, temperature 20 °C. The monitoring wavelength was 280 nm and 320. Calibration curves were performed between 20–50 µg/mL. Results were expressed in mg/L.

### 3.7. Microbiological Analysis

Total plate counts (TPC) and *Enterobacteriaceae* count (ENT) were carried out to evaluate the microbiological quality of the samples. The analyses were performed in triplicate according to the methodology established by the American Public Health Association. The samples were collected aseptically in a sterile vessel, immediately after processing. For sample preparation, mixing was performed using a stirrer to ensure a uniform distribution of the microorganisms in the mass product, followed by five dilutions consecutively (10–5). As dilution liquid was used sterile distilled water. TPC determination was performed according to SR EN ISO 4833 [56] using Plate Count Agar (PCA) medium and the detection of *Enterobacteriaceae* was performed according to ISO 21528-1,2, [57] using VRBG medium (agar with purple red ball glucose). The PCA medium allows the growth of all aerobic germs and the other two culture media have selective character, containing inhibitory factors for the development of other microorganisms, other than *Enterobacteriaceae*, respectively yeasts and molds. After sowing of two Petri plates from each consecutive serial dilution, they were incubated for a different period of time, namely: 72 h for TPC at 30 °C and 37 °C and 24 h in the case of *Enterobacteriaceae* at 37 °C [58]. The results were expressed as log colony forming units per milliliter of juice (log CFU/mL).

### 3.8. Data Analysis

All values were obtained from three independent experiments and each red grape juice sample was analyzed in triplicate. All obtained results are presented as average values followed by the standard deviation (SD) of three replicates. The one-way analysis of variance (ANOVA) test was used to analyze the data and differences among means obtained in response to the applied treatments were compared by a Tukey test with a level of significance of *p* < 0.05. The statistical data analysis was performed using the JASP (Version 0.11.1, 2019) computer software (JASP Team, University of Amsterdam, Amsterdam, The Netherlandss).

## 4. Conclusions

The obtained results reveal that compared to pasteurization, the HPU treatment significantly improved the individual polyphenolic compounds and the L-AsAc content, without any significant effect on pH, TA and TSS. Also, important decreases in microbial counts without affecting the investigated bioactive compounds and physicochemical parameters of grape juice samples were induced by HPU treatment, compared to pasteurization. The TPC and ENT analysis results of the sonicated samples were significantly influenced by treatment time and amplitude level. The quality of red grape juice is influenced by choosing appropriate ultrasound treatment parameters. The ultrasound treatments with an amplitude of 70% were more efficient than those with A50% in terms of all investigated parameters. As for the processing time, our results reveal that while HPU treatment (A70%, 5 min) is recommended for enhancing the level of flavonoids such as rutin, quercetin, epicatechin, kaempferol as well as resveratrol, the HPU treatment (A70%, 10 min) was the most efficient in the reduction of the microbial load and obtaining the highest content of L-AsAc and TP together with a high level of some hydroxycinnamic acids as ferulic and caffeic acid. The data reported in this study suggest that HPU treatment might be used as an alternative emerging technique to successfully replace the pasteurization for improving the quality of the red grape juice. Nevertheless, we could go further in the experiments to establish the most adequate operating conditions of HPU treatment to be in agreement with national and international requirements in terms of microbial load for extending the self-life of grape juice.

## Figures and Tables

**Figure 1 molecules-25-01669-f001:**
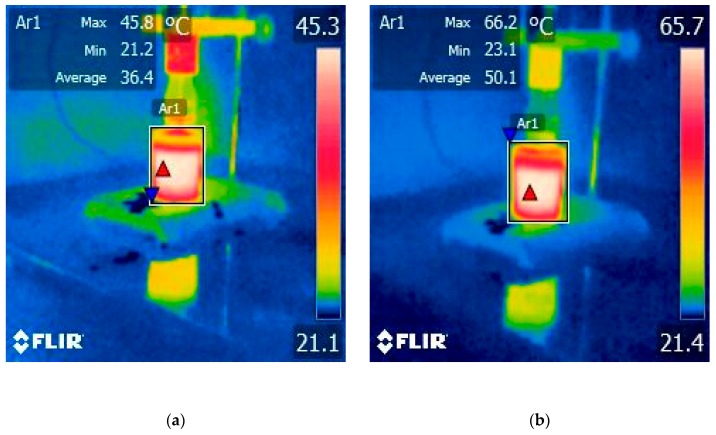
The temperature measured after 5 min of HPU treatment: (**a**) A50%; (**b**) A70%.

**Figure 2 molecules-25-01669-f002:**
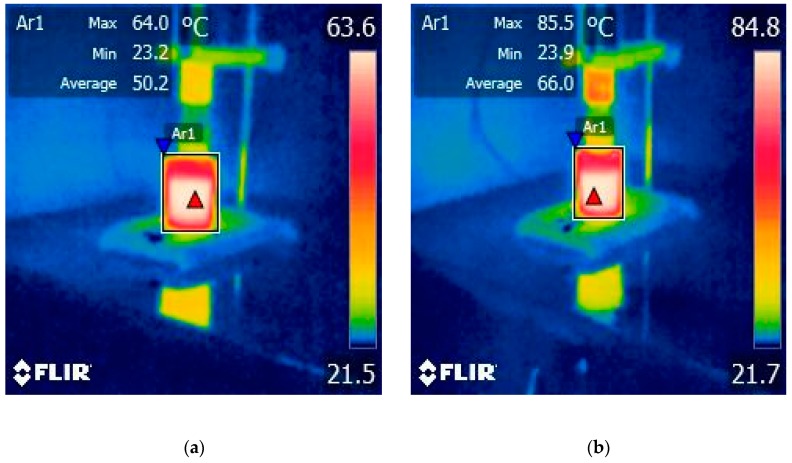
The temperature measured after 10 min of HPU treatment: (**a**) A50%; (**b**) A70%.

**Figure 3 molecules-25-01669-f003:**
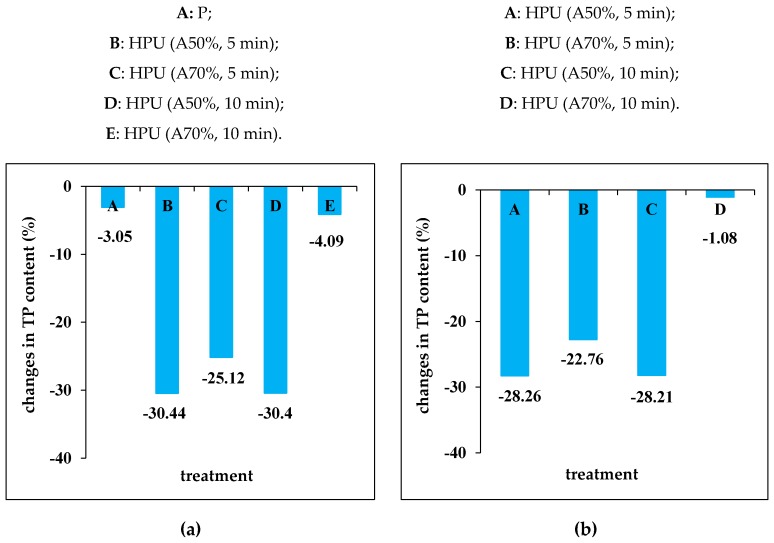
The changes in total phenolic (TP) content of juice samples in response to the applied treatments: (**a**) reported to the control sample (C); (**b**) reported to the pasteurized sample (P).

**Figure 4 molecules-25-01669-f004:**
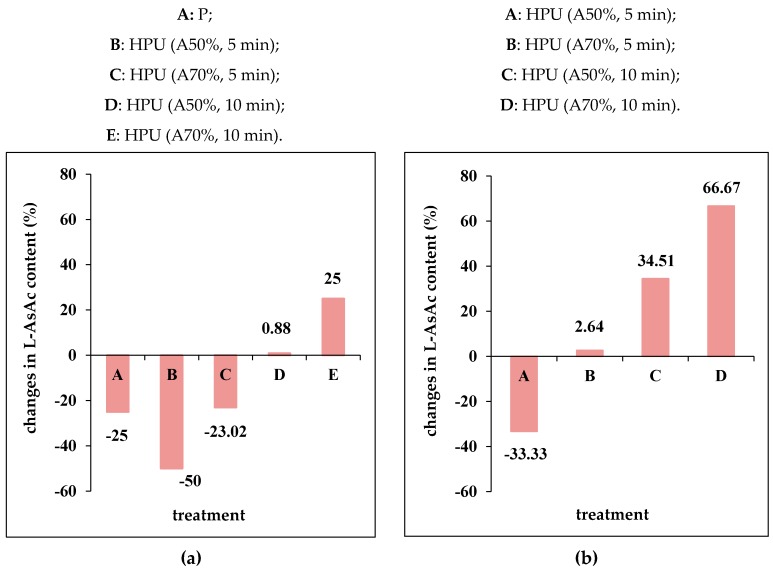
The changes in l-ascorbic acid (L-AsAc) content of juice samples in response to the applied treatments: (**a**) reported to the control sample (C); (**b**) reported to the pasteurized sample (P).

**Table 1 molecules-25-01669-t001:** Changes in temperature of juice samples as a result of ultrasound treatments parameters.

**Time of Temperature Measuring (min)**	**T(°C)**	
	**HPU (A50%, 5 min)**	**HPU (A70%, 5 min)**
0	18.9 ± 0.85	19.7 ± 0.70
2.5	25.7 ± 0.90	36.9 ± 1.00
5	36.4 ± 1.20	50.1 ± 0.70
**Time of Temperature Measuring (min)**	**T(°C)**	
	**HPU (A50%, 10 min)**	**HPU (A70%, 10 min)**
0	17.2 ± 0.85	19.9 ± 0.20
2.5	25.8 ± 0.70	34.9 ± 1.00
5	33.8 ± 1.30	51.4 ± 0.90
10	50.2 ± 1.20	66.0 ± 1.10

Results are expressed as the average value of three replicates ± standard deviation (SD).

**Table 2 molecules-25-01669-t002:** Effect of pasteurization and HPU treatments on the total phenolic (TP) and l-ascorbic acid (L-AsAc) content.

Parameter	Grape Juice Sample					
	C*	P**	HPU (A50%, 5 min)	HPU (A70%, 5 min)	HPU (A50%, 10 min)	HPU (A70%, 10 min)
TP (mg GAE/ 100 mL)	72.86 ± 0.02^a^	70.64 ± 0.08^b^	50.68 ± 0.05^c^	54.56 ± 0.07^d^	50.71 ± 0.09^c^	69.88 ± 0.02^e^
L-AsAc (mg/L)	454.4 ± 0.11^a^	340.8 ± 0.09^b^	227.2 ± 0.04^c^	349.8 ± 0.05^b^	458.4 ± 0.10^a^	568 ± 0.08^d^

* Control sample; ** pasteurized juice sample. One-way ANOVA test was used to compare the means differences among treatments; different superscripts in the same row indicate significant differences among the treatments (Tukey’s test, *p* < 0.05). Results are expressed as the average value of three replicates ± standard deviation (SD).

**Table 3 molecules-25-01669-t003:** Effect of pasteurization and HPU treatments on the polyphenolic compounds profile of grape juice samples.

Polyphenolic Compounds (mg/L)	Grape Juice Sample					
	C*	P**	HPU (A50%, 5 min)	HPU (A70%, 5 min)	HPU (A50%, 10 min)	HPU (A70%, 10 min)
Gallic acid	2.15 ± 0.03^a^	2.16 ± 0.05^a^	15.23 ± 0.04^b^	31.30 ± 0.07^c^	12.82 ± 0.11^b^	19.49 ± 0.09^c^
Protocatechuic acid	n.d.	2.08 ± 0.35^a^	n.d.	5.13 ± 0.05 ^b^	0.79 ± 0.44^c^	n.d.
Caffeic acid	0.93 ± 0.21^a^	2.62 ± 0.37^a^	1.30 ± 0.52^a^	14.31 ± 0.06^ab^	4.41 ± 0.08^ab^	23.27 ± 0.04^ab^
Epicatechin	3.70 ± 0.16^a^	10.70 ± 0.33^a^	5.45 ± 0.15^a^	76.80 ± 0.07^ab^	n.d.	35.31 ± 0.54^ab^
p-cumaric acid	0.28 ± 0.03^a^	0.1 ± 0.11^a^	0.04 ± 0.04^a^	1.20 ± 0.04^a^	1.13 ± 0.05^a^	n.d.
Ferulic acid	0.27 ± 0,01^a^	0.32 ± 0.03^a^	n.d.	n.d.	n.d.	0.81 ± 0.12^b^
Rutin	2.09 ± 0.11^a^	2.51 ± 0.06^a^	9.61 ± 0.1^a^	63.31 ± 0.02^b^	n.d.	39.02 ± 0.1^ab^
Rosmarinic acid	n.d.	n.d.	1.19 ± 0.06^a^	25.93 ± 0.21^b^	0.39 ± 0.08^a^	4.31 ± 0.05^b^
Resveratrol	0.21 ± 0.12^a^	n.d.	0.65 ± 0.13^ab^	8.54 ± 0.14^b^	0.80 ± 0.06^ab^	2.48 ± 0.04^b^
Quercetin	0.38 ± 0.07^a^	n.d.	n.d.	5.62 ± 0.02^b^	0.31 ± 0.08^a^	0.71 ± 0.03^ab^
Kaempferol	0.83 ± 0.05^a^	0.92 ± 0.06^a^	n.d.	4.59 ± 0.12^b^	n.d.	n.d.

* Control sample; ** pasteurized juice sample; n.d. - not detected. One-way ANOVA test was used to compare the means differences among treatments; different superscripts in the same row indicate significant differences among the treatments (Tukey’s test, *p* < 0.05). Results are expressed as the average value of three replicates ± standard deviation (SD).

**Table 4 molecules-25-01669-t004:** Effects of ultrasound and pasteurization treatments on the pH, total soluble solids (TSS) and titratable acidity (TA).

Parameter	Grape Juice Sample					
	C*	P**	HPU (A50%, 5 min)	HPU (A70%, 5 min)	HPU (A50%, 10 min)	HPU (A70%, 10 min)
pH	3.62 ± 0.01^a^	3.59 ± 0.01^b^	3.52 ± 0.01^c^	3.51 ± 0.01^c^	3.50 ± 0.01^d^	3.50 ± 0.01 ^d^
TSS (°Brix)	30.18 ± 0.13^a^	30.73 ± 0.09^a^	26.47 ± 0.09^bc^	27.73 ± 0.09^b^	27.13 ± 0,05^abc^	29.67 ± 0.09^ab^
TA (g of tartaric acid /L)	3.75 ± 0.07^a^	4.23 ± 0.08^b^	3.79 ± 0,03^a^	3.97 ± 0.12^abc^	3.94 ± 0.16^ab^	4.18 ± 0.11^bc^

* Control sample; ** pasteurized juice sample. One-way ANOVA test was used to compare the means differences among treatments; different superscripts in the same row indicate significant differences among the treatments (Tukey’s test, *p* < 0.05). Results are expressed as the average value of three replicates ± standard deviation (SD).

**Table 5 molecules-25-01669-t005:** Impact of pasteurization and HPU treatments on the total plate count (TPC) and the *Enterobacteriaceae* count (ENT) of grape juice.

Parameter	Grape Juice Sample					
	C*	P**	HPU (A50%, 5 min)	HPU (A70%, 5 min)	HPU (A50%, 10 min)	HPU (A70%, 10 min)
TPC (log CFU/mL)	4.53 ± 0.01^a^	2.53 ± 0.07^b^	2.18 ± 0.03^cd^	1.42 ± 0.02^cd^	1.63 ± 0.02^c^	1.13 ± 0.04^c^
ENT (log CFU/mL)	1.51 ± 0.01^a^	0.73 ± 0.04^ab^	1.45 ± 0.05^ab^	0.69 ± 0.03^ab^	1.27 ± 0.02^acd^	0.53 ± 0.0^acd^

* Control sample; ** pasteurized juice sample. One-way ANOVA test was used to compare the means differences among treatments; different superscripts in the same row indicate significant differences among the treatments (Tukey’s test, *p* < 0.05). Results are expressed as the average value of three replicates ± standard deviation (SD).

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
