# Peer review of "An Overview of Effects Induced by Pasteurization and High-Power Ultrasound Treatment on the Quality of Red Grape Juice"

_molecules, 2020, doi:10.3390/molecules25071669_

Round 1

Reviewer 1 Report

The manuscript is well written.
The methods are suitable and all applied methods
are necessary for adequate explanation of results.
Also, obtained results are of great importance for
any production of fruit juices after high-power ultrasound treatment.

Reviewer 2 Report

See pdf adjunto

Reviewer 3 Report

In description of tables and figures please use the full words and then abbreviation in brackets.

For example: not only TPC

but: Total plate counts (TPC)

and so on.

In Materials and methods section please describe in a couple sentences what amplitude in HPU is.

line 275: have, not has

line 311: established, not establishing : reduction, not reducing

Round 2

Reviewer 2 Report

The manuscript has been significantly improved.

Therefore, I suggest its publication in Molecules.